

# High microphone signal-to-noise ratio enhances acoustic sampling of wildlife

Kevin F.A. Darras[1], Franziska Deppe[1], Yvonne Fabian[1,5], Agus P. Kartono[2], Andres Angulo[1], Bjørn Kolbrek[3], Yeni A. Mulyani[2] and Dewi M. Prawiradilaga[4]

[1] Agroecology, University of Göttingen, Göttingen, Niedersachsen, Germany
[2] Department of Forest Resources, Conservation and Ecotourism, Bogor Institute of Agriculture, Bogor, Indonesia
[3] Celestion International, Ipswich, Suffolk, United Kingdom
[4] Museum Zoologicum Bogoriense Research Centre for Biology-LIPI Jl, Bogor, Indonesia
[5] Agroscope FAL Reckenholz, Swiss Federal Research Station for Agroecology and Agriculture, Züich, Switzerland

## ABSTRACT

**Background**. Automated sound recorders are a popular sampling tool in ecology. However, the microphones themselves received little attention so far, and specifications that determine the recordings' sound quality are seldom mentioned. Here, we demonstrate the importance of microphone signal-to-noise ratio for sampling sonant animals.

**Methods**. We tested 12 different microphone models in the field and measured their signal-to-noise ratios and detection ranges. We also measured the vocalisation activity of birds and bats that they recorded, the bird species richness, the bat call types richness, as well as the performance of automated detection of bird and bat calls. We tested the relationship of each one of these measures with signal-to-noise ratio in statistical models.

**Results**. Microphone signal-to-noise ratio positively affects the sound detection space areas, which increased by a factor of 1.7 for audible sound, and 10 for ultrasound, from the lowest to the highest signal-to-noise ratio microphone. Consequently, the sampled vocalisation activity increased by a factor of 1.6 for birds, and 9.7 for bats. Correspondingly, the species pool of birds and bats could not be completely detected by the microphones with lower signal-to-noise ratio. The performance of automated detection of bird and bat calls, as measured by its precision and recall, increased significantly with microphone signal-to-noise ratio.

**Discussion**. Microphone signal-to-noise ratio is a crucial characteristic of a sound recording system, positively affecting the acoustic sampling performance of birds and bats. It should be maximised by choosing appropriate microphones, and be quantified independently, especially in the ultrasound range.

Corresponding author
Kevin F.A. Darras, kdarras@gwdg.de

# INTRODUCTION

Acoustic recording of wildlife is a popular sampling method for sonant animals such as birds and bats (*Gibb et al., 2018*; *Sugai et al., 2019*). The ecologist can control the sound

recording quality by choosing high-quality microphones, ensuring that the animal sounds of interest can be detected (*Fristrup & Mennitt, 2012*). Microphone quality is commonly described by its self-noise and signal-to-noise ratio (SNR). Self-noise is the noise produced by the microphone in the absence of sound, and is typically given in dB SPL (decibel sound pressure level, defined as 20 times the logarithm of the ratio of the sound pressure to the reference sound pressure of 20 μPa) A-weighted. It describes the equivalent background noise level that would be measured by a perfect (noiseless) microphone, and is ideally measured by placing the microphone in a sound-proof container. Microphone self-noise defines the lowest sound pressure level the microphone can detect, and also the resulting SNR of the recorded signals. Signal-to-noise ratio in dB is defined as 10 times the logarithm of ratio of a standard signal's power to the noise power of the microphone created by its self-noise (*Kinsler et al., 1999*; *International Organization for Standardization, 2019*):

$$SNR = 10 \times \left( \frac{Power_{signal}}{Power_{noise}} \right)$$

The standard signal is commonly generated by a sound calibrator with a 94 dB SPL tone at a 1 kHz sound frequency. Signal-to-noise ratio is a relative measure, valid only for a given signal level, while self-noise is an absolute measure of the microphone quality. Signal-to-noise ratio at a calibrated SPL will however give a measure of self-noise, because it is obtained by subtracting the self-noise from the standard signal's level.

In the following, we will focus on the more commonly mentioned SNR rather than the microphone self-noise. Its importance is routinely implied in technical literature about microphones (e.g., *Lewis & Schreier, 2013*). In contrast, in ecoacoustics, little attention has been paid to the microphone specifications. Some studies have evaluated the effectiveness of different recorder types for birds (*Rempel et al., 2013*; *Pérez-Granados et al., 2019a*; *Pérez-Granados et al., 2019b*) and bats (*Adams et al., 2012*), and it has been reported that a recording system with the lowest SNR detected the least birds (*Rempel et al., 2013*). Finally, *Bardeli et al. (2010)* mentioned that automated detection of animal sounds could be impeded by worn microphones. However, out of 20 published studies used in a recent meta-analysis about autonomous sound recording (*Darras et al., 2018a*), only six mentioned SNR, and only two of those specified the SNR of their own microphones. However, it is the first element in the signal processing chain and it determines the output recording's quality.

Technical specifications of microphones, including their SNR, have an impact on the sampling effectiveness, probably through their impact on the detection ranges: microphones that have a low SNR (a high self-noise) add too much noise to the recordings, so that the animal sounds—especially faint, distant ones—are not detectable anymore (*Darras et al., 2018a*). However, it is crucial to maximise detection ranges to cover the largest possible sampling areas in little time. So far, an experimental proof of the relationship between SNR and detection ranges is still missing, although SNR is a technical measure that we can control by design, in contrast to the other biotic and abiotic factors that determine sound detection spaces (*Darras et al., 2016*). Moreover, since high SNR leads to less noise in recordings, the recordings would be easier to analyse for human listeners. By extension, high

SNR also facilitates the automated detection of animal sounds (*Jančovič & Köküer, 2011*), as well as their classification (*Chen & Maher, 2006*), both of which are commonly used in acoustic bat surveys, and increasingly used for bird surveys (*Priyadarshani, Marsland & Castro, 2018*).

We aim to understand how microphone SNR affects the recording of soundscapes for ecologists sampling sonant biodiversity. We used 12 different microphone models spanning a wide range of SNR values to evaluate how it affects acoustic sampling of birds and bats, the most frequently acoustically sampled animals. In the field, we recorded silence to determine the self-noise floor of the microphones, sound transmission sequences to determine the microphones' SNR and their detection ranges, and bats at night and birds during the morning to determine their sampling efficiency. We measured the microphones' calibrated SNR values and compared them to manufacturer specifications to check their reliability; for the first time, we measured microphone SNR in the ultrasound range. We tested whether SNR determines the detection ranges of the microphones at audible and ultrasonic frequencies. Since detection ranges determine the acoustic sampling areas, we ultimately tested whether SNR affects the measured activity and richness of the sampled birds and bats. Additionally, we tested the relationship between SNR and the precision and recall of and bird and bat call automated detection.

## MATERIALS AND METHODS

### Microphones

We used omnidirectional microphone elements of different types and models to test a wide range of manufacturer-specified SNR values that are typical for soundscape recording applications. We used two units for each of 12 different models from six different manufacturers, with specified SNR at 1 kHz from 55 to 80 dB, resulting in a total of 24 microphone elements (Table 1). Most of the chosen microphones are used in commercial or open-source microphones for soundscape recording. Eight microphone element models were in the form of traditional cylindrical capsules, and two of them are used in commercial microphones (WM-61A in SMX-II, FG-23629-C36 in SMX-U1, Wildlife acoustics). The AOM-5024L-HD-R is interchangeable with the EM172 element used in the Solo open-source recording system (*Whytock & Christie, 2017*). Four models were Micro-Electro-Mechanical Systems (hereafter MEMS) chips that can be integrated on printed circuit boards, two of which were used in commercial products (SPU0410LR5H-QB in Bio-SMX-US, Biotope.fr, SPM0404UD5 in SMX-US, Wildlife acoustics). Four of the MEMS elements and one capsule microphone element are also part of the open-source microphone system for ecoacoustics "Sonitor" (*Darras et al., 2018b*). Finally, to extend the range of tested SNR values, we included a microphone with a lower SNR (POM-1345P-C3310-R, 55 dB) that is not used in microphones for soundscape recording. The prices per unit at purchase ranged from approximately 0.5 EUR (typical for MEMS elements) to 4 EUR (typical for high-end capsules), depending on the ordered quantity, with the notable exception of the FG Knowles element, which cost 23 EUR. Details about microphone assembly are in the supplementary materials.
**Table 1  Microphone elements used in the study, along with letter codes used in Fig. 1.**

| Microphone element | Code | Type | Format | Manufacturer | Signal-to-noise ratio 1kHz (dB) | Sensitivity (dB SPL) |
|---|---|---|---|---|---|---|
| POM-1345P-C3310-R | A | | | | 55 | −45 |
| POM-2735P-R | B | | | | 60 | −35 |
| ROM-2235P-HD-R | C | | capsule | PUI Audio | 68 | −35 |
| POM-2730L-HD-R | D | electret condenser | | | 74 | −30 |
| AOM-5024L-HD-R | E | | | | 80 | −24 |
| ICS-40720 | F | | MEMS | Invensense | 70 | −38 |
| WM-61A | G | | capsule | Panasonic | 62 | −35 |
| PMM-3738-VM1000-R | H | piezoelectric | | Vesper | 62 | −38 |
| SPM0404UD5 | I | | MEMS | | 59 | −42 |
| SPU0410LR5H-QB | J | electret condenser | | Knowles | 63 | −38 |
| FG-23629-C36 | K | | capsule | | 66 | −53 |
| EM258 | L | | | Primo | 74 | −32 |

**Notes.**

MEMS, Microelectromechanical systems.

Signal-to-noise ratios and sensitivity values provided by manufacturer.

Our microphones were calibrated at 1 and 40 kHz using reference microphones for audible sound calibration, because not all had the same, standard format to fit a sound calibrator. For audible sound, we used the SMX-US (discontinued - Wildlife Acoustics, Massachusetts, USA): it has a standard 1/2 inch diameter, fitting into a class I sound calibrator (PCE-SC42, PCE instruments, Germany) that emits a 94 dB SPL tone of 1 kHz. For ultrasound, we used the reference microphone ICS-40720; it was calibrated with the ultrasound calibrator (Wildlife Acoustics, Massachusetts, USA) that emits a 48 dB SPL tone of 40 kHz in "chirp" mode (measured at a distance of 30 cm, because microphones were not plugged into it).

## Study site and setup

We set up the microphones in a research plot situated in an oil palm plantation (S01.70725, E103.39781, WGS84 datum) belonging to the PTPN6 state-owned company in Sumatra, Indonesia. Our research group was part of the CRC990 project, which has a Memorandum of Understanding with PTPN6. We installed 12 microphones—one of each model—simultaneously in a microphone holder, to equalise the soundscapes that they record. The holder consisted of a wooden pane (Fig. S1, approximately 25 cm × 35 cm, installed at a height of 1.5 m) fitted with synthetic foam strips to absorb sound reflections. The holes where the microphones were inserted were also padded with foam to avoid friction noise. The microphones were positioned horizontally and arranged in a regular grid, with three microphones in each of four rows, spaced 10 cm vertically and 15 cm horizontally. We removed wind screens, which incur a slight loss in sound transmission (<1 dB according to Wildlife Acoustics), to measure the performance of the actual microphone elements. We recorded on rain- and wind-free days, as most microphones were not protected. The microphones were connected to six sound recorders (SM2Bat+, Wildlife Acoustics) with 5 m cables. We chose the SM2Bat+ because it was the only recorder that we possessed which

was compatible with all microphones. The manufacturer specifies an electric noise floor of -115 dB V for 44.1 kHz recordings and -105 dB V for 192 kHz recordings. After completing all measurements with the first set of microphones, we repeated them with the second set on the following day, placing the second unit of each model in the same position as with the previous set.

## Sound level measurement

We recorded all sound in uncompressed audio (WAV format) in stereo (2 channels), sampled at 192 kHz (16 bit per sample) in November 2018. We visualized, processed, and measured audio in Audacity (Audacity Team, 2018). We generated their spectrograms with a Fast-Fourier-Transform with a Hanning window size of 1024. For measuring sound levels, we re-sampled audio at a sampling rate of 96 kHz to obtain a higher frequency resolution in the spectrograms. We measured relative sound levels in dB in the frequency bins containing the 1 kHz and at 40 kHz test frequencies using the ''Plot spectrum'' function. We used a 0.8 s sound selection for audible signals and 0.07 s for the short ultrasound chirps. We chose to consistently use the 1024 window size as it is the default setting in Audacity that allowed the best trade-off between temporal and frequency resolutions for locating and measuring the short 40 kHz signal tone in space and time. Since we are dealing with field measurements with noise outside of the test frequencies, we did not use root-mean-squared values to measure sound pressure levels as they would cover the entire frequency spectrum. Moreover, we needed to apply filtering in the following for aurally and visually detecting those signals, justifying the use of sound levels derived from specific frequency bins.

## Self-noise

We measured microphone self-noise by recording sound in an environment that was as silent as possible. We did not have access to an anechoic chamber and preferred to record silence in the oil palm plantation, far from anthropogenic machinery noise rather than in the laboratory or other urban buildings. We used an isolating, large cylindrical ice box with a hole in its cover to pass the microphone cables. The box was padded inside with synthetic foam and surrounded by a thick polyester sleeping bag to prevent extraneous noise from reaching the microphones, resulting in a basic anechoic box. We started the recording on all recorders, knocked on the box to be able to synchronize them later, and recorded silence for one minute at a 96 kHz sampling rate. We measured the relative sound pressure levels in dB inside the silent recording for all microphones at 1 and 40 kHz. We used the same simultaneous 60 s of sound for all microphones of one set.

## Sound transmission sequences

We recorded sound transmission sequences (*Darras et al., 2016*) to determine relative microphone sensitivity, to compute SNR, and to measure detection spaces. We generated an audio recording in Audacity, consisting of a sequence of 1 s long test tones at 1 kHz, repeated at five different sound levels (see Supplementary materials) to be able to choose the most appropriate sound level *a posteriori*. Audible test sounds came from a battery-powered one driver loudspeaker (SoundCore Anker) at default loudness (the level automatically reset to when powering on), with the audio recording loaded on a Mini-SD card. We emitted

ultrasound with the ultrasonic calibrator, which produces 40 kHz chirps of constant loudness in "Chirp" mode. Sound transmission sequences are obtained by recording test tones emitted at different distances from the microphone: The loudspeaker and ultrasound calibrator were held at a height of 2 m and pointed to the microphones as they emitted test sounds at distances of 2, 4, 6, 8, 10, 15, 20, 25, 30, 35, 40, 45, and 50 m to the front, the left, the right, and the back of the microphones to determine the entire sound detection area. At 1 and 40 kHz, we used the nearest common distance at which none of the microphones recorded clipped (saturated) test tones to measure their sensitivity relative to the ones recorded by the calibrated reference microphone; these tones had the same source sound pressure level and distance to all microphones.

## Sound detection spaces

For measuring the microphones' detection ranges, we chose the loudest of the five recorded sound levels at which none of the microphones recorded detectable (i.e., not visible on spectrogram and not audible in recording) test tones at 50 m by focusing on the relevant frequency (1 or 40 kHz). By doing this, we ensured that we measured maximal detection ranges for each microphone, and obtained the most accurate relative range differences. We determined the detection range as extinction distances: the distance at which the test tone was not detectable anymore because it equals the ambient sound level (*Darras et al., 2016*). Since above 10 m, we only emitted test tones every 5 m, a single listener (FD) estimated the detection range to the meter based on how loud the last detectable test sound was (visually assessed from the spectrogram). For some combinations of microphone and direction however, detection ranges exceeded 50 m at 1 kHz, so the listener estimated the extrapolated detection range above 50 m.

The detection ranges in the four different directions formed four quarter-ellipses which were used to calculate the sound detection space areas (using coord_polar function of ggplot2). We did not use the mean detection ranges in our analysis because our setup could have resulted in directional sound pickup patterns, and also because detection area is ultimately the measure that determines the sampling area for our organisms of interest. The sound detection spaces we measured represent standardised detection ranges that are specific to the particular sounds we emitted; they are characterised by their frequency, the emitting hardware's source level, directivity, height, orientation, and the habitat of the recording site (*Darras et al., 2016*). This is in contrast to effective detection radii: these depend on the organisms' vocalising frequency, amplitude, orientation, and directivity, and they represent maximum distances up to which specific species can be recorded with a particular sound recording system and habitat (*Matsuoka et al., 2012*).

## Bird and bat activity and richness

After carrying out the sound transmission recordings with each set, the microphones were left in place to record sound on the following night (for bats) and morning (for birds). We used a sampling rate of 44.1 kHz during the day for birds and the usual 192 kHz sampling rate for bats at night. After retrieval, the recordings' sound levels were normalised (Normalize function in Audacity).

We uploaded normalised recordings of the first 30 min after sunrise for birds and the first 30 min after sunset for bats to our online platform for ecoacoustics BioSounds and analysed them to quantify vocalising activities of bird and bat species (*Darras et al., 2020*). We screened synchronous recordings from every microphone, opened side-by-side in different internet browser tabs. The recordings' spectrograms and audio enabled visual and aural filtering and detection of bird and bat vocalisations by FD. The recordings that appeared to have the best quality (clearest spectrogram and noise-free audio) were used as reference (Knowles FG-23629-C36 for bats, Primo EM258 for birds) as clear differences were visible in the spectrograms between microphones (Fig. S2). Whenever a vocalisation was found in the reference recording, it was annotated to extract its coordinates in time. Calls of the same species that were separated by less than 15 (for birds) or 5 s (for bats) were put in a common tag to ensure standardised tagging among recordings; tagging every call is too tedious and does not reflect the activity of birds singing at large intervals accurately. Note that different thresholds could result in slightly different results and that different tags could belong to different or identical individuals. However, it is more important to use a consistent threshold across the variables of interest (here, the microphones) for unbiased results. All other non-reference recordings were checked at the same time point and if vocalisations could be found, they were also tagged. We checked whether other vocalisations were missed in the non-reference recordings that could not be found in the reference recording.

We measured bird and bat activity and richness. AA manually assigned bird vocalisations to species using Birdlife International taxonomy, and FD manually assigned bat vocalisations to call types due to the lack of comprehensive reference libraries for South-East Asia, and also because bat species identity *per se* does not affect our results. We counted richness as the number of bird species or bat call types for each microphone. We computed the total duration of tagged vocalisations for birds and bats for each microphone, which yielded the vocalisation activity, in seconds.

## Automated detection of calls

We carried out automated detection of bird and bat calls with "acoustic recognizers". We exported the same, loudest bird and bat call sequence (to ensure that they are recorded by all microphones) of the most common species from each microphone's unamplified recording. Along with these call sequences, we exported two representative calls contained within them, which were also the same across microphones. We used the latter as templates to detect all other calls in the exported bird and bat call sequences using spectrogram cross-correlation with monitoR (*Katz, Hafner & Donovan, 2016*). We ordered the detections made by monitoR by descending score and AA manually assigned them to true and false positives by visually checking their spectrograms. After twenty consecutive false positives, we deemed all the following detections to be false positives also. We counted the actual number of calls (equal to the sum of true positives and false negatives) visually in the spectrogram of the bird and bat call sequences. Bat calls from the second night were distant, so that they were not picked up by three microphones.

## Statistical analysis and graphing

Statistical tests were performed in R 3.6.1 (*R Core Team, 2019*); plots were drawn using ggplot2 (*Wickham, 2009*); the code and output can be checked in the HTML report (Supplementary Materials). Actual effect sizes for the detection area and vocalisation activity models are hard to interpret due to different predictor transformations and the use of different generalised model families; they are also specific to our study site, so we report the ranges of the response variables instead.

## Measured vs. specified signal-to-noise

The relative sound pressure levels output by the recorder were standardised by subtracting the amplification applied by the recorder. We then computed the SNR of each microphone by subtracting its relative sensitivity difference and its self-noise from the sound pressure level of the calibration tone recorded by the reference microphone. The obtained SNR is relative to the absolute sound pressure level of the calibrators (94 dB SPL for 1 kHz, 48 dB SPL for 40 kHz).

We plotted the measured and specified (i.e., manufacturer given) microphone SNR to check how consistent they are across manufacturers. Note that we did not expect our SNR values to equal manufacturer specifications due to the different procedures used for measuring them: we use frequency-specific measures of sound level while manufacturers use broad-band root-mean-squared measures of sound level. However, we expected that similar specified SNR values across microphones of different manufacturers yield similar measured SNR values. Some microphones had acoustic vents (GAW112, Gore, USA) in front of them, which reduce sound transmission by <1 dB (specified by the manufacturer) while providing protection against water ingress; we corrected this by adding 1 dB to their measured SNR, only when comparing them to manufacturer SNRs. We compared the corrected Akaike Information Criterion (Sugiura 1978) of linear models predicting the measured SNR using all combinations of the continuous specified SNR and the categorical manufacturer predictor: the null model without predictors, a simple model with specified SNR only, one with an additional manufacturer-specific offset, and one with the additional interaction between manufacturer and specified SNR. We used the model with the lowest AICc—and consequently highest predictive power—to draw regression lines of predicted values of measured and specified SNR and checked the diagnostic plots of its simulated residuals (DHARMa package in R).

## Signal-to-noise vs. detection space area

We tested for the relationship between microphone SNR and sound detection space area, depending on the frequency, using a linear regression model. Depending on the range of SNR values under consideration, SNR may have an accelerating effect on detection spaces at low SNR values (a linearly increasing range leads to a quadratic increase in the detection circle's area), a roughly linear effect at intermediate SNR values, and a decelerating effect at high SNR values, where the detection area is increasingly limited by the ambient sound (*Darras et al., 2018a*). As a consequence, we tested three models: one with a linear SNR predictor, one with a second-order polynomial of the SNR predictor (for modeling convex

and concave relationships), and one with a log-transformed SNR predictor (which has one more degree of freedom than the second-order polynomial, but cannot model an increasing slope). We compared the AICc value of our models to choose the one with the highest predictive power and checked its simulated residuals. Note that we did not model more complex relationships between SNR and detection area because a multitude of interacting, and sometimes opposed factors affect sound transmission (*Darras et al., 2016*).

## Signal-to-noise vs. vocalization activity

We tested the relationship between microphone SNR and bird and bat activity using a model with the sampling day as an additional fixed effect to account for day-to-day variations in animal activity. We chose the best SNR predictor transformation as determined by the analysis of detection spaces, corresponding to the frequency of interest. For the bird and bat activity datasets, we tested different model types to obtain the best fit. We tested a linear model that accounts for the fact that at high values, the activity distribution can tend to normality (Central limit theorem); we tested a generalised model with Poisson family to account for the fact that activities are derived from a count of animals; we tested a model with a negative-binomial family to account for overdispersion, and finally, we tested a model with Gamma family to account for the fact that activities are positive and continuous values. We compared the AICc value of our models to choose the one with the highest predictive power and checked its simulated residuals.

## Signal-to-noise vs. species richness

The sampled species richness can be affected by microphone SNR either by the speed at which the maximum number of sampled species is reached, or by the maximum level of sampled species itself. We counted the number of sampled species at 40 time steps along the 30 min of recording, to graphically depict the species pool and the speed at which it is sampled with time for microphones with different SNR. We refrained from analysing the effect of SNR on species richness with statistical models for the following reasons: (1) oil palm plantations have limited species pools: in our study area, we expect no more than four call types for echolocating bats, and seven bird species (*Darras et al., 2019*), so that we only have a very limited range for the response variable; (2) typical count-variable models (such as Poisson or negative binomial) are modeling unbounded count response variables, and as such inappropriate; (3) the statistical analysis of the effect of SNR could be carried out at each of the different time steps at which the species richness is counted, but count-variable models would not converge at early time steps when the response variable comprises mostly zeroes, at intermediate time steps, the results are driven by outliers, and at late time steps, we are essentially only testing the effect of SNRs on the maximal level of the species pool only, which is very limited; (4) modeling the species accumulation with time using saturating curves with a Michaelis–Menten dynamic yields too many unrealistic outliers that do not correspond to the maximal number of species present in oil palm plantations.

**Table 2** Results for selecting the best model predicting measured signal-to-noise ratio with specified signal-to-noise ratio based on corrected Akaike Information Criterion.

| Model | Degrees of freedom | AICc | Formula (SNR in dB @ 1 kHz) |
|---|---|---|---|
| Null | 2 | 182.43 | $\text{SNR}_{measured} \sim 1$ |
| Simple | 3 | 176.06 | $\text{SNR}_{measured} \sim \text{SNR}_{specified}$ |
| Manufacturer | 8 | 159.12 | $\text{SNR}_{measured} \sim \text{SNR}_{specified} + \text{Manufacturer}$ |
| Full | 9 | 165.04 | $\text{SNR}_{measured} \sim \text{SNR}_{specified} * \text{Manufacturer}$ |

## Signal-to-noise vs. automated detection performance

We analysed the influence of the microphone SNR on the performance of the automated detection algorithm used in monitoR, which matches a template with each time bin in the recording (*Katz, Hafner & Donovan, 2016*). We follow the recommendations of *Knight et al. (2017)* to assess the performance of acoustic automated detection algorithms (i.e., acoustic recognizers) with the precision—which describes the proportion of detections that are true positives—and the recall—which describes the proportion of the real number of calls that are detected by the recognizer. In line with their recommendations, we account for the influence of the threshold (or cutoff) of the score, which measures the level of confidence in the detection, used for classifying audio events as detections or not, which ranges from 0 (no match with template) to 1 (perfect match). We depicted the total number of available calls along with the true and false positives, in relation to the score cutoff used to count them. We defined minimal precision and recall levels of 0.5 as acceptable and computed the range of score cutoff values over which we would obtain an acceptable automated detection performance. Finally, we analysed the effect of SNR on the acceptable range of score cutoffs with a linear model, using the sampling day and frequency as fixed effects, along with the interaction between SNR and frequency, and we checked its simulated residuals.

# RESULTS

## Measured vs. specified signal-to-noise

The linear model using specified SNR and manufacturer-specific intercepts as predictors had the highest predictive power (i.e., lowest AICc: Table 2; Fig. 1) and the simulated residuals showed no deviations (HTML report, supplementary materials). We detected large differences between manufacturers: for instance, we measured SNR values for PUI Audio that were on average 26 dB below those we measured for Knowles microphone elements.

## Signal-to-noise vs. detection space area

The model with log-transformed SNR that predicted detection areas at 1 kHz had the lowest AICc, while the model with non-transformed SNR that predicted detection areas at 40 kHz had the lowest AICc (Table 3). The simulated residuals of the chosen models showed no deviations (HTML report, supplementary materials). Detection areas increased significantly with SNR from 207 to 10,477 $m^2$ at 1 kHz and from 353 to 3632 $m^2$ at 40 kHz

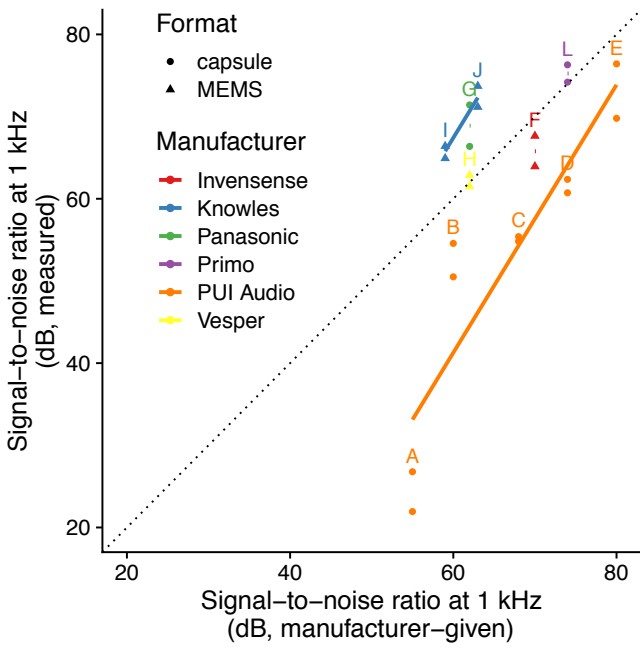

**Figure 1** **Relationship between specified and measured microphone signal-to-noise ratio at 1 kHz for different manufacturers and formats.** Letters identify microphone models (see Table 1). The dotted line indicates a 1:1 relationship, and solid lines indicate linear fits for each manufacturer with more than one microphone manufacturer. Microphone self-noise measurement protocols differ between measured and specified values.

**Table 3** **Results for selecting the best model predicting detection spaces with measured signal-to-noise ratio based on corrected Akaike Information Criterion.**

| Frequency | Model | Degrees of freedom | AICc | Formula (area in m², SNR in dB) |
|---|---|---|---|---|
| | Linear | 3 | 367.18 | Detection area ∼SNR |
| 1 kHz | Log-linear | 3 | 363.77 | Detection area ∼log(SNR) |
| | Polynomial | 4 | 364.89 | Detection area ∼poly(SNR, 2) |
| | Linear | 3 | 354.07 | Detection area ∼SNR |
| 40 kHz | Log-linear | 3 | 355.81 | Detection area ∼log(SNR) |
| | Polynomial | 4 | 356.21 | Detection area ∼poly(SNR, 2) |

(all $P < 0.001$, adjusted $R^2$ at 1 kHz: 0.91; at 40 kHz: 0.70; Fig. 2). Detection ranges roughly corresponded to beams of omni-directional microphones (Fig. S3).

## Signal-to-noise vs. vocalisation activity

We detected nine bird species (*Chalcophaps indica*, *Dicaeum trigonostigma*, *Geopelia striata*, *Halcyon smyrnensis*, *Orthotomus atrogularis*, *O. ruficeps*, *O. sericeus*, *Pycnonotus aurigaster*, *P. goiavier*) and four bat call types (A-D). The linear model that predicted bird activities with log-transformed SNR had the lowest AICc, and the generalised negative binomial linear model that predicted bat activities with SNR had the lowest AICc (Table 4). The simulated residuals of the chosen models showed no strong deviations (HTML report,

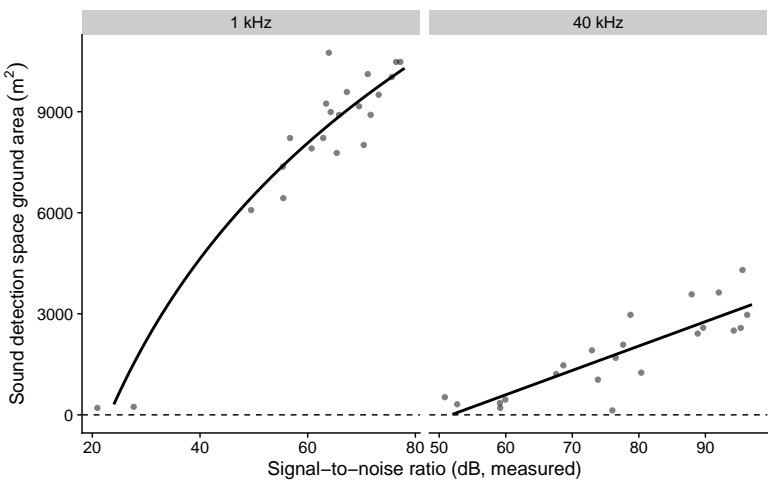

**Figure 2** **The influence of measured microphone signal-to-noise ratio on detection space areas in the audible (1 kHz) and ultrasound (40 kHz) ranges.** Lines show predictions from a linear regression against log-transformed (for birds) and non-transformed signal-to-noise ratio (for bats).

**Table 4** **Results for selecting the best model predicting bird and bat vocalisation activities with measured signal-to-noise ratio based on corrected Akaike Information Criterion.** We did not test a linear model for bats because the data were distinctly count-distributed. We included the interaction with the recording nights for bat models due to large differences between recording nights.

| Taxon | Model | Degrees of freedom | AICc | Formula activity: min for birds, s for bats |
|-------|-------|--------------------|------|---------------------------------------------|
| Birds | linear | 4 | 95.64 | total_activity ~log(SNR_dB) + recording_date |
| | GLM | 3 | 122.27 | round(total_activity) ~log(SNR_dB) + recording_date |
| | GLM negative binomial | 4 | 125.35 | round(total_activity) ~log(SNR_dB) + recording_date |
| | GLM Gamma | 4 | 156.74 | total_activity + 0.01 ~log(SNR_dB) + recording_date |
| Bats | GLM | 4 | 338.79 | round(total_activity) ~SNR_dB * recording.date |
| | GLM negative binomial | 5 | 227.86 | round(total_activity) ~SNR_dB * recording.date |
| | GLM Gamma | 5 | 235.11 | total_activity ~SNR_dB * recording.date |

supplementary materials). The sampled vocalisation activities significantly increased with SNR (Fig. 3). During the day with most animal activity, activities ranged from 950 to 1539 s for birds ($P < 0.001$, adjusted $R^2$: 0.92, using only microphones commonly employed for ecoacoustics), and from 33 to 320 s for bats ($P < 0.001$, adjusted $R^2$: 0.90). The microphone model with the lowest SNR (POM-1345P-C3310-R) did not pick up any bird calls.

## Signal-to-noise vs. species richness
Maximal bat species richness levels generally reached higher levels, at a higher rate, with increasing microphone SNR (Fig. 4). For birds, we could only observe higher maximal sampled richness levels with SNR for one of the two sampling days (5th of November).

## Signal-to-noise vs. automated detection performance
The performance of automated detection of bird and bat calls was positively affected by microphone SNR (Fig. 5). The simulated residuals showed no deviations (HTML report,

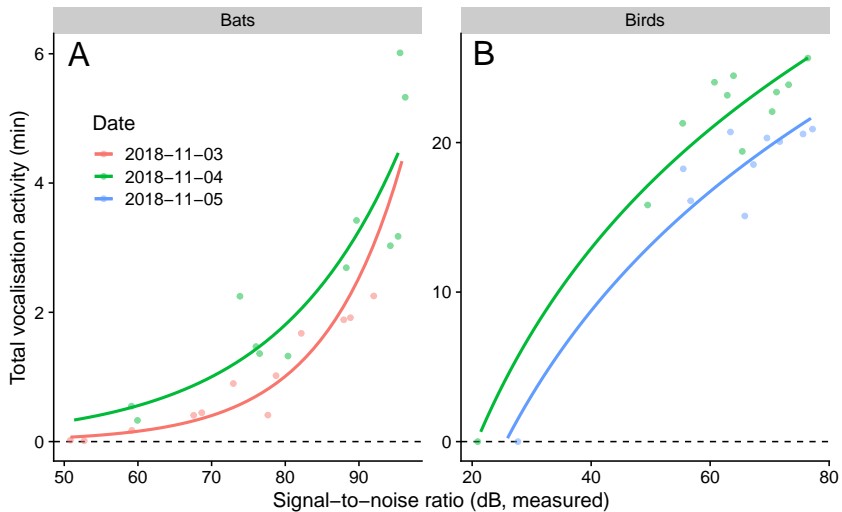

**Figure 3** **Vocalisation activity of birds and bats against microphone signal-to-noise ratio in the audible (1 kHz for birds) and ultrasound (40 kHz for bats) ranges.** Lines show predictions from a generalised linear model with a negative binomial family (for bats) and a linear model using log-transformed signal-to-noise ratio (for birds). Sampling day was used as fixed effect.

supplementary materials). The range of score cutoff values that had acceptable recall and precision (i.e., a minimum of 0.5) significantly increased by 0.9 and 0.4 per SNR unit increase, respectively for bird and bat calls (all $P > 0.001$, adjusted $R^2$: 0.80).

# DISCUSSION

Our measured SNR values showed large discrepancies between microphone element manufacturers. Increasing signal-to-noise ratios considerably enlarged sound detection areas for audible sound and ultrasound. In turn, the sampled bird and bat activity and richness was largely enhanced by high microphone SNR, and automated detection performance of bat calls was facilitated by high microphone SNR.

## Measuring signal-to-noise ratios and detection ranges

For ecoacoustic studies, we need standardised microphone SNR values for specific frequencies of interest corresponding to different animal groups. Although we only measured two frequencies, SNR is a continuous measure that varies with frequency, and depending on the targeted organism, other frequencies (or frequency bands) will be more relevant to determine the fit of different microphones. Microphone manufacturer-provided SNR did not correlate well with our standardised SNR measurements because of strong differences between manufacturers. Microphone element manufacturers do not follow a standard certification for measuring microphone SNR (pers. comm. PUI audio). Moreover, SNR values are usually only specified for 1 kHz, a representative frequency for human speech (but see *Knowles, 2014*), so that we could not compare our signal-to-noise measurements at 40 kHz with any reference. However, for bat researchers, we showed that it is important to have high-SNR microphones, but this cannot be achieved without
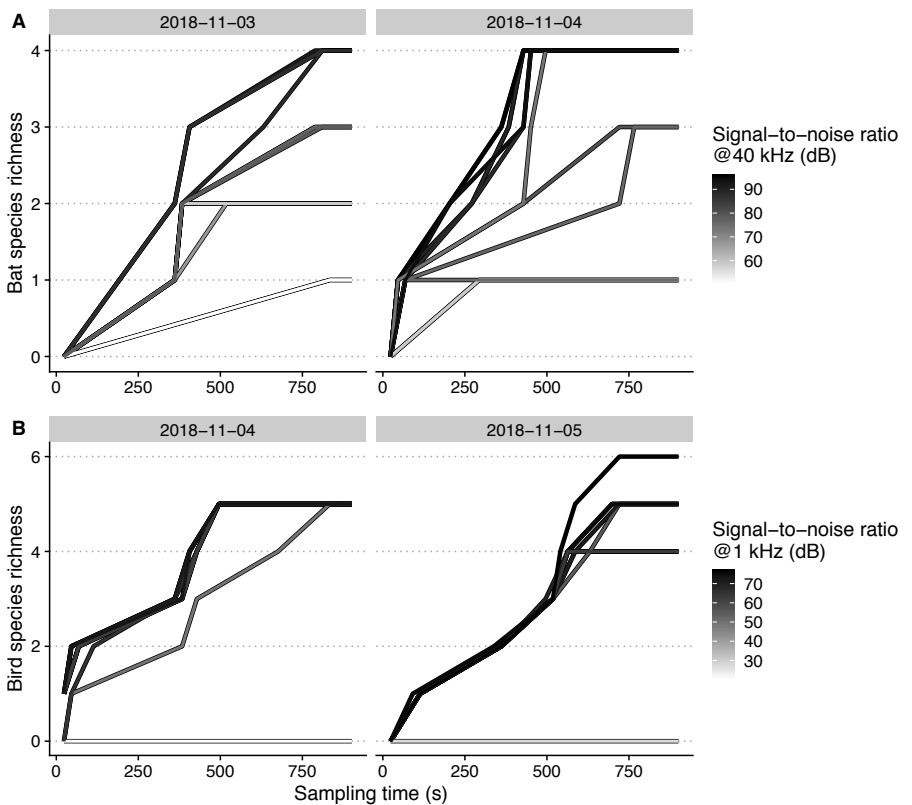

**Figure 4 Species accumulation curves of each microphone, recording bat (A) and bird (B) species on different days, plotted against sampling time.** Signal-to-noise ratios are scaled within each taxon for achieving higher color contrast.

knowing their ultrasonic SNR beforehand. Furthermore, SNR values at 1 kHz are only weakly indicative of microphone performance in the ultrasound range (Fig. S4), even though some of the variation in our ultrasound SNR values might have been caused by variable alignment of the microphones. Finally, our self-noise measurements were carried out for specific frequencies, but manufacturers usually measure signal-to-noise—which is based on the self-noise measure—with A-weighting (which weights human-audible frequencies more) over a 20 Hz to 20 kHz bandwidth that encompasses different groups of vocalising animals. Moreover, different weightings and bandwidths can lead to different SNR values. In future studies, SNR measurements in audible sound and ultrasound ranges should be compiled by researchers with standard measurement protocols for different microphone models to support microphone selection for ecoacoustic studies. Note that as SNR varies with frequency, it can be derived from the frequency and noise responses of the microphones to frequency by subtraction, for a more complete assessment of microphone quality across the entire acoustic frequency range. Unfortunately, such data are rarely available (but see *Knowles, 2014*).

Using our sound detection area measurement approach, one could also directly benchmark microphones based on their detection ranges. Even though signal-to-noise

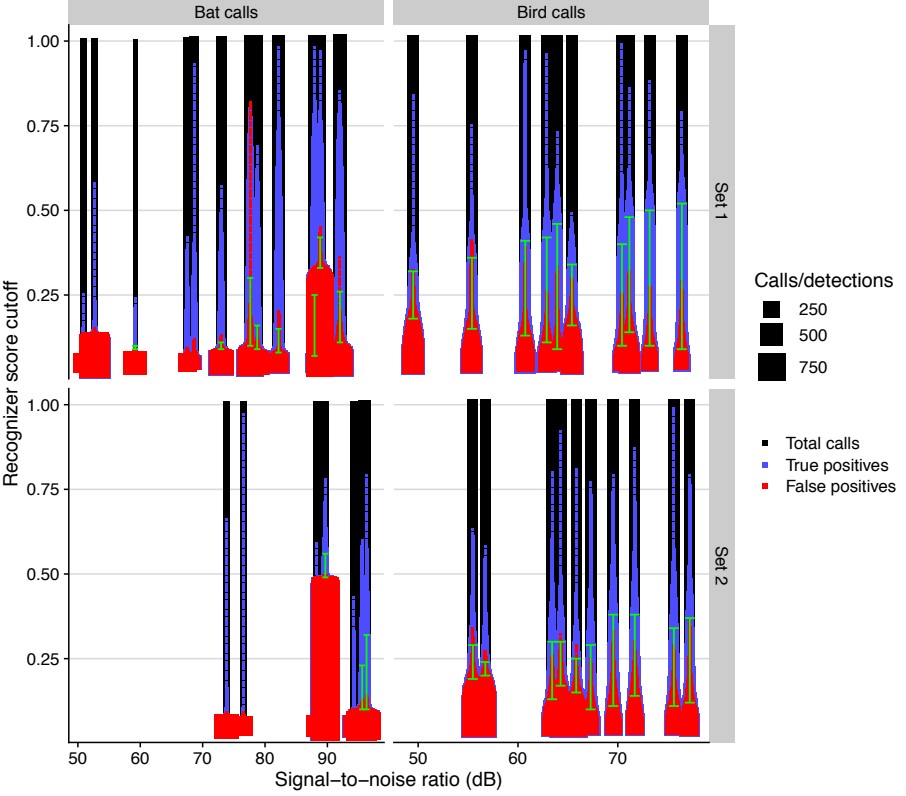

**Figure 5** **Automated call detection performance for birds and bats for different days and microphone sets, measured by the total number of calls, true, and false positives, depending on recognizer score cutoff.** We defined acceptable precision and recall values with a value of 0.5 and show the range of recognizer score cutoffs where these minimal values are attained with green bars.

ratio seems to be the main determinant of sound detection spaces (and is more easily measured), other microphone characteristics like linearity and directionality theoretically also affect them. However, all the microphones we used were specified as linear and omnidirectional by the manufacturers, and any possible deviation from this was assumed to be random across microphone models. It would be possible to devise protocols to measure the sound detection spaces of sound recording setups - recorders with microphones, or only microphones - of different manufacturers to assess their sampling effectiveness in a standardised way. Relative differences should be independent of the habitat in which recordings are made, but absolute ranges would vary between habitats (*Darras et al., 2016*). Possibly, results of previously published comparisons of bird and bat recorders (*Adams et al., 2012*; *Rempel et al., 2013*), which did not explicitly consider microphones, could be explained quantitatively by simple differences in detection ranges caused by differing microphone SNR values.

We consider that by definition, sensitivity is secondary compared to self-noise or SNR values with respect to their impact on detection ranges, in slight contrast to previous findings about the importance of microphone sensitivity (*Turgeon, Van Wilgenburg &*

*Drake, 2017*). Microphones used for wildlife recordings usually have a sensitivity of -36 dBV, their levels are equalised with amplification, and the added noise that arises from this amplification is generally negligible: Even with our discontinued recorders, and only at ultrasound sampling frequencies, only the ICS-40720 and Primo EM258 microphone elements are noticeably limited by the recorder by having higher electrical noise floors than the amplifier. Thus, we argue that for a broad range of commercially available microphone elements that come into question for wildlife recordings, SNR values are the limiting factor determining sound detection areas.

## Maximising recording quality

The inherent sampling effectiveness of sound recorders, in terms of sampling area, should be maximised by choosing microphones with high SNR values. In our case, we could reach half of the largest detection area with SNRs as low as 42 dB for birds and 81 dB for bats. However, this number depends on the range of SNR values covered by our selected microphones. In contrast, a previous meta-analytical approach showed that microphones for audible sound perform as well as human observers at SNR values of approximately 80 dB (*Darras et al., 2018a*). Thus, more importantly, detectability increases with SNR: even though extremely low SNR microphone elements are almost non-existent, it is worthwhile to search for the highest SNR in the market of existing microphone elements. Moreover, the detection areas of our wide range of microphone elements did not reach a clear saturation point, although detection spaces are eventually limited by the ambient sound (*Apol, Valentine & Proppe, 2019*; *Darras et al., 2018a*). Crucially, the best-performance microphones can be obtained at little additional expense: as an example, the lowest signal-to-noise PUI Audio microphone element (55 dB SPL) cost us 3.10 EUR (at the time of purchase, via Digikey.com), only a little less (28%) than the highest signal-to-noise PUI Audio microphone element (80 dB SPL) that cost 4.25 EUR. Compared to the total cost of microphones, these expenses are negligible, and even more so with even cheaper MEMS microphones, which can cost roughly half as much.

Low signal-to-noise microphones are not unusable, but they come with manageable drawbacks. Even though it is more efficient to sample birds and bats with high-quality ratio microphones, in some special cases where the sampling area should not be too large (e.g., when sampling needs to be limited to small habitat patches), having small microphone detection ranges can be an advantage. However, artificially adding noise to high-quality recordings could be a more flexible approach and have the same effect (*Darras et al., 2016*). Also, sub-optimal recordings from low-quality microphones can be used too, as long as longer recording durations are used, to measure additively higher vocalisation activities and to reach higher species richness as shown by our species accumulation curves (Fig. 3). Low-quality microphone recordings can also be used together with high-quality recordings when accounting explicitly for the different detectabilities (e.g., with occupancy modeling approaches), or for the relationship between vocal activity rate and bird abundance (*Pérez-Granados et al., 2019a*; *Pérez-Granados et al., 2019b*).

For species richness, the outcome of increasing SNR and sampling durations may depend on the animals' mobility. For mobile organisms like bats that are constantly roaming the

study site, the entire species pool could be sampled eventually—even when detection spaces are small because of low-quality microphones—given enough sampling time. For territorial organisms like birds, that have perching preferences and mostly vocalize when perching, the species pool could remain incompletely sampled even after long recording durations. However, our results contradicted our expectation, and we recommend more in-depth studies in habitats with larger species pools, sampling for longer durations.

Still, high SNR values are required for accurate localisation of sound sources (*Good & Gilkey, 1996*), which would support the estimation of bird detection distances for distance sampling (*Darras et al., 2018c*). In practice, recordings from microphones with high SNR also show clearer signals with higher SNR (Fig. S5). We showed how this facilitates automated detection: microphones with low SNR do not have an acceptable performance in terms of precision and recall. Analogously to sound recordings, the lower the SNR, the lower the signal of detection events and the higher the signal of the background noise, to the point that it is impossible to distinguish true detection events from spurious detection events found by the recognizer inside ambient noise, leading to low precision. Correspondingly, the lower the SNR, the narrower the range of score thresholds wherein automated detection performs satisfyingly, and in the worst case, automated detection becomes impossible. It is also conceivable that analogously, high SNR values facilitate visual and aural inspection of sound recordings by humans.

Finally, for all microphones, sensitivity and SNR values degrade with time, so that they should be regularly assessed to keep sampling effectiveness to a maximum or to account for their variable detection ranges (*Darras et al., 2018a*; *Turgeon, Van Wilgenburg & Drake, 2017*). However, even with a technically high-quality sound recording setup, the deployment should still follow general recommendations for soundscape recordings (*Darras et al., 2018a*), such as installing microphones above the lower vegetation layer, spacing them and pointing them in opposite directions, and using lossless compressed audio at high sampling rates.

## CONCLUSIONS

We suggest that ecologists recognize microphone SNR or self-noise as a standard metric for assessing microphone quality in ecoacoustics. Microphone SNR largely determines the sound detection space. Through this, it dictates how many individuals are recorded and how efficiently species pools can be sampled. High microphone SNR values also enhance the precision and recall of automatic detection methods for bird and bat calls. Thus, high-quality microphones are paramount for achieving maximum detection ranges with accurately detectable sounds.

## ACKNOWLEDGEMENTS

We would like to offer our most sincere condolences to the family of Agus Priyono Kartono, who passed away during the study. We thank the reviewers for their extremely valuable help in improving this paper. We are grateful to our counterpart Damayanti Buchori for supporting this research project.

### Funding

This study was financed by the Deutsche Forschungsgemeinschaft (DFG) in the framework of the collaborative German—Indonesian research project CRC990. We received support from the German Research Foundation and the Open Access Publication Fund of the Göttingen University. There was no additional external funding received for this study. The funders had no role in study design, data collection and analysis, decision to publish, or preparation of the manuscript.

### Grant Disclosures

The following grant information was disclosed by the authors:
Deutsche Forschungsgemeinschaft (DFG).
German Research Foundation.
Open Access Publication Fund of the Göttingen University.

### Competing Interests

The authors declare there are no competing interests. Bjørn Kolbrek is employed by Celestion International.

### Author Contributions

- Kevin F.A. Darras conceived and designed the experiments, performed the experiments, analyzed the data, prepared figures and/or tables, authored or reviewed drafts of the paper, and approved the final draft.
- Franziska Deppe and Yvonne Fabian performed the experiments, authored or reviewed drafts of the paper, and approved the final draft.
- Agus P. Kartono, Andres Angulo, Yeni A. Mulyani and Dewi M. Prawiradilaga analyzed the data, authored or reviewed drafts of the paper, and approved the final draft.
- Bjørn Kolbrek analyzed the data, authored or reviewed drafts of the paper, acoustics expertise, and approved the final draft.

### Field Study Permissions

The following information was supplied relating to field study approvals (i.e., approving body and any reference numbers):

A field permit was not required as our research group was part of the CRC990 project, which has a formal, general MoU with the owner of the plantation, PTPN6.

### Data Availability

Raw data and R script are available in the Supplemental Files.

Data are contained within CSV files and results, models and graphs can be reproduced by running the R script.

## Supplemental Information

Supplemental information for this article can be found online at http://dx.doi.org/10.7717/peerj.9955#supplemental-information.

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
