# Peer review of "High microphone signal-to-noise ratio enhances acoustic sampling of wildlife"

_PeerJ, doi:10.7717/peerj.9955_

## Round 0.1 · original submission · Major Revisions

Thank you for your contribution. I was able to secure three good and very detailed reviewers all specialists in the field. Reviewer #1 suggested a couple of references that deserve to be added to the text. I second him on this and also have a couple of comments on that matter, see comments in the PDF. The Methods section needs to be much better organized, data analysis is somewhat intermixed with other methods. Please, separate that on its own subheading. Reviewer #2 suggested a number of aspects of the study that can be addressed in the discussion to improve it. I highly recommend you to take a careful look at those and consider talking about them in the discussion. Lastly, all three reviewers pointed out that the Methods section needs to include many more details. This critique also applies to the Abstract, the Methods section needs to include analysis at least.
Since you run all analysis in R, I highly suggest you provide an R Markdown dynamic document with the workflow of analysis as a supplemental file.
Remember to reply to my comments from the PDF in your rebuttal letter

·

Basic reporting

I think that it is a very interesting paper filling a clear gap in the literature. The authors did a really great job and I would like to congratulate them. I have just two major comments and a list of minor comments that I hope may help to improve the current version of the manuscript, which I think that is worthy of publication once these issues have been solved.

I think that there adequate references not included in the text that need to be included and what is more important considered them to improve the Introduction about what is already known and improve the discussion of your results. These references may also help to support some of the assumptions or definitions made by the authors that are currently no supported for references. I feel that a carefull reading of the following references (and cite when appropiate) may improve your text and provide more info to readers interested on further information.
a) In the Fristrup and Mennit (2012) and Obris et al. (2010) works there are a section dedicated to microphones and explaining some of the reasoning and definitions included in your reference. They could be a good starting point for readers interested on Reading more about this topic.
Obrist MK, Pavan G, Sueur J, Riede K, Llusia D, Márquez R. (2010). Bioacoustics approaches in biodiversity inventories. Abc Taxa 8:68-99.
Fristrup, K. M., and D. Mennitt. 2012. Bioacoustical monitoring in terrestrial environments. Acoustics Today 8(3):16-24.
b) There are two studies that compared the effectiveness of recorders for bird monitoring that should be read and cited in the text (see also minor comments). Among them it could more interesting to read carefully the one carried out by Rempel et al. (2013), since they make references to the SNR of the recorders tested.
Rempel RS, Francis CM, Robinson JN, Campbell M. (2013). Comparison of audio
recording system performance for detecting and monitoring songbirds. J Field Ornithol 84:86-97.
Pérez-Granados C, Bota G, Giralt D, Albarracín J, Traba J. (2019). Cost-effective assessment of five audio recording systems for wildlife monitoring: differences between recording distances and singing direction. Ardeola 66:311-325.
c) The study of Turgeon et al. (2017) although cited in the text I also believe that that deserves more attention. For example they compared the SNR of the microphones that they used with the ones given by the manufacturer and some of their statements are very similar to some of yours. I quote a few examples:
“Furthermore, we show that lower microphone sensitivity reduces the effective area sampled by a microphone and thus induces distance-related biases in detection probability for all species”. It is in agreement with your results, and might be nice to say something in the Introduction.
“Furthermore, automated detection or recognition software will likely be more affected by worn microphones compared with human analysts because noise impedes the ability of automated recognizers to detect sounds (Bardeli et al. 2010).” Check the Bardeli reference but this assumption migth be also considered.
“Performing a single point check using a commercially available 1 KHz/94 dB sound level calibrator should be sufficient to identify microphones with poor sensitivity because sensitivity loss appears to be greatest in the low frequencies.” Maybe some discussion with your results could be started from here.

There some other references that I feel adequate included in the minor comments, but I think that a carefull read of these five references might be interesting before starting the review.

Experimental design

The section dedicated to statistical methods needs to be improved. This is a section that deserves more attention to understand the results and have some chances of replicate the analyses or make future comparisons. I think that you should create a subheading where the readers may look into what analyses you did. I have added some minor comments about it when reviewing the text, but the main point is that the methods applied are not described or described with little detail.

Validity of the findings

No comment. I trust on the results although an enlarged explanation of the methods employed (statistical methods) is needed.

I would like to ask to the authors about why they did not test for differences in quality among the six recorders employed in the study. They found differences between the manufacturer specifications and the “real” ones for the microphones, so why they did not assess for diferences in the recorders employed? I truly believe in your results but I hope you see my point.

Additional comments

Minor comments.

Introduction:
L. 44. I feel that the reference to Darras et al. 2019 should be changed for a more appropiate one (e.g. Sugai, L. S. M., Silva, T. S. F., Ribeiro Jr, J. W., & Llusia, D. (2019). Terrestrial passive acoustic monitoring: review and perspectives. BioScience, 69(1), 15-25).
L. 46. I see the point, but microphone quality is not the only factor that determines whether the recordings quality is sufficient to detect animal sounds. There other factors, such as background noise, distance of the vocalizing animal, singing direction, other taxa vocalizing at the same time, etc. I totally understand what the authors want to say, but try to clarify.
L. 48-61. Maybe try to add some adequate references about SNR or SPL. Likely most of the sentences of the text have been read from some study, cite some general studies with these definitions (e.g. SPL as 20 times the logarithm of ratio of the sound pressure to…. or SNR as the 10 times the logarithm of ratio of a standard’s signal). See some of my “general references”.
L. 65: Please, replace this reference of Venier for the two more appropiate testing the utility of different recorders for bird monitoring (Rempel et al. 2013 and Pérez-Granados et al. 2019). These authors tested the effectiveness of a large number of recorders under different circumstances, such as distance or singing direction or devices cost. These two references are a better background for readers interested on the topic. Maybe the Venier reference does not need to be removed if the authors consider it interesting, but add the other two.
L. 67. Remove space after “2018”. And valorate if the term “For example” might me better than “Indeed”.
L. 74. As I said before valorate if the results of the Rempel study are worthy to be cited here too.
L. 76-77. Are you sure that the Kaplan reference is valid for the statement that “high signal-to-noise ratios should facilitate the automated detection of animal sounds”? It is only that I look too old and for sure automated detection have improved a lot since this year and what he said may not be longer na issue. Even, after reading the study I am not sure if Kapplan applied automated detection.
L. 78. Why bat surveys? I guess that this assumption should be valid for all monitored taxa, and this work is not focused on bats.
L. 81. Use a different term than “our”.
L. 83. “their” sampling efficiency?
L. 86. We tested (in past)
L. 90. Maybe use something like “we assessed whether” is more adequate than “we demonstrate”.
Material and methods
L. 96. When I read the abstract (“We tested 24 different microphones”) I expected to find a comparison of 24 different microphones. However, you assessed “only” 12 microphones (two replicates of each). The study is still nice and interesting, but clarify that you tested 12 types of microfones and not 24 different microphones in the abstract and throughout the text.
L. 96. I think the term “quality” is subjective (high, low, medium?), maybe the term settings/configurations or other similar is more adequate.
L. 117-118. How much was that lost? You are doing an statement without any reference, so I have to assume that it is based on personal observation (although not said). This lost is interesting to make comparisons on further studies (thay may choose for removing or not the wind screens) or to calibrate them on future research, since probably most of future surveys aiming to perform long-term surveys (one of the main advantages of acoustic monitoring) will use the microphones with wind screens.
L. 123-124. The different types of microphones were tested in the same position in the recorders? I mean, the first unit of microphone “A” was tested on the right channel of the recorder “A”. Then, the second unit of the microphone type “A” was also placed in the right channel of the recorder “A” or it was random?
L. 136. There are pictures of how silent recordings were recorded? It would be nice to add this picture as Supplemental or even in the main text, since it may help readers to understand the methodology applied.
L. 183-186. But what was the analyses? Which software and function did you use? What was the variable response and the predictor? Did you check the model? Much more info is needed.
L. 215: Cite some studies showing these effects of habitat or singing directions.
L. 215-218. I think that this section deserves more attention. You should clearly clarify that you are making a specific test for your study, but that the detection ranges provided here are not, are may not be, equal to the effective detection radius. I know it and I understand what you did, but I think that it is easy that some readers may use your detection areas as effective detection radius if using the same microphone. There are studies showing that the Song Meter-II recorder (that likely used some of the microphones tested in your study) was able to effectively record bird vocalizations at a distances further than 150 m (see Rempel et al. 2013, Turgeon et al. 2017 and Pérez-Granados et al. 2019).
L. 220-223: Only one round means 12 microphones? Provide info about recording Schedule, format of recording, sample and bit rate, date, etc.
L. 233. Calls less than 15 s for the same bird? It is not possible that two birds vocalize in the same 15 s? Please, clarify this selection which seems arbitrary.
L. 244-246. It is impossible to valorate the statistical analyses section. There is no info about the recording schedule, which function in R was applied (please cite the packages used too), how was the model evaluated, etc. Looking to the graph 2 it seems that you used day as random factor besides you only had three monitoring days. If that is the case this statistical analyses might not be adequate since using a random factor with such small number of levels is not a good approach (a minimum number of five levels is recommended). I consider to reading more about that and if I am correct using day as factor in a GLM. Likely, your results will be pretty the same.
L. 264. Please cite some previous studies that have used the same approach and defined this term. I also consider that you should use the term that is usually used when testing the recognizer performance (recall rate), since it may help readers to understand what you did. (Knight, E., Hannah, K., Foley, G., Scott, C., Brigham, R., & Bayne, E. (2017). Recommendations for acoustic recognizer performance assessment with application to five common automated signal recognition programs. Avian Conservation and Ecology, 12(2):14)
L. 267-269. More info is required to understand what are “external” templates.

Results:
I think that this section could be improved by adding some table or figure summarizing the main results. The section is very nice and well written but sometimes just difficult to follow. Maybe you could add in Table 1 the SNR of the microphones provided by the manufacturer and the ones measured in the field.
Again the section dedicated to external detection templates was difficult to understand. Maybe it is my fault, of course, but please, try to clarify this section in methods and results to avoid confussion.
L. 296-299. I think that these sentences should be moved to methods.

Discussion:
I think that could be also interesting to discuss something about how most of previous studies focused their main goals in comparing recorders but they did not pay attention to microphones, besides likely they are more important than the recording unit itself. In this sense your study is filling a clear gap.
L. 349. In the Rempel et al. 2013 study they find almost the same and included a picture of the sonogram of the same call recorded at different distances to see the diferences between recorders, making some reference about their SNR. Please, check it.
L. 353.355. Almost the same. I think that this statement that you are doing here is pretty the same that Pérez-Granados et al. 2019 already did using also automated recognition software. When focusing in the tests carried out at favourable singing direction and the SM2 recorder (likely very similar to the SM2Bat of your study) they found that the total number and percentage of bird songs recognized in respect to total songs uttered by playback was almost the double when using the own recording of the SM2 that when using the calls broadcast by the playback (material appendix). I think that it means the same that what you are saying here (it is better to use the own recording collected by the recorder that using reference material). In that case could be nice to support your results in this prior work.
L. 377. I guess you mean “total cost of recorders”.
L. 383-384. “Also, sub-optimal recordings from low-quality microphones can be used too by sampling for longer durations to obtain higher vocalisation activities”. How it works? I guess that the mean parameters that alter how long you can record are sample rate and format (mp3, wav, wac). However, I cannot see how using a microphone with different SRN may make you record for longer durations.
L. 384: Valorate if saying “the results may depend.
L. 394-395. Order references by time. With these prices (less than 5 E per microphone), I think that a recommendation could be just to replace microphones at least once per year. This is not money for most projects, for the price of just one SM2Bat unit you can replace hundreds of microphones.

Conclusions:
L. 397-398. I think that SNR is already a standard metric for assessing microphone quality. The problem is that very little attention have been paid to measure the SNR of the microphones or at least make a small test of the microphones/recorders to be used. I think that as a conclusion of the study you should say something about the need to describe the type and the SNR of microphones employed in future studies on a similar way that almost every study describe the sampling and bit rate employed for recording. It will facilitate future comparisons and provide more date to undertand the results obtained in future studies. As guidance, Turgeon et al. (2017) said “It would allow the inclusion of microphone sensitivity as a covariate in statistical models to potentially adjust for differences in detection between sites or years”.

Abstract:
L. 25. You should say that you tested 12 type of microphones (add if you want that two replicates per test).
L. 27-29. I think that the interesting point is not that you measured these variables (other studies did before), the interesting point is that you assessed the relationship between SNR of the microphones and these variables.

Table 1: The SNL and SPL provided are the ones provided by the manufacturer? Please, clarify.
Fig. 1. It would be very nice to see the effect of the singing direction on the sound detection space by showing the results of your playbacks from different sides of the recorder on a 3D or circular graph. This new graph may help to understand how the sound detection space varies with the location of vocalizing animals, which may have great consequences about how placing the recorders in future surveys.
Fig. 4. This result is amazing and clearly suggest thet need to estimate the recall rate of the recognizers employed on every single study, even if using a small proportion of the recordings since the recall rate may greatly differ with the microphone employed. Thanks, very interesting. Assess if including something about it in the Discussion.

Supplemental Figures: There are lot of figures that are not cited in the text. I am not aware of the PeerJ but likely citing Figure S4 but not the previous is not correct. Likewise, I think that it could be interesting to upload all these figures in TIF, PNG or JPEG, since likely there will be lot of readers not able to open EPS figures.

References:
This section needs to be checked carefully. There are plenty of references with different style. In some of them (e.g. Gibb et al. ) the year of the publication is not provided, in others the authors cite et al. when there are several authors while in others add “…”. Likewise, in some references (e.g. Turgeon et al.) the number of the article is not provided. For some references a letter after the year should be added (e.g. there is Darras et al. 2019b but not Darras et al. 2019a).

Thanks for this work,
Cristian Pérez-Granados.

Reviewer 2 ·

Basic reporting

This manuscript report how distinct signal-to-noise ratios (SNR) of microphones can potentially impact biodiversity acoustic monitoring. As open-source platforms of sensor for biodiversity monitoring are increasingly being adopted in ecological research, this contribution is useful to guide choosing acoustic sensors based on their specifications. It is clear that the authors have put a lot of effort into this work and I commend their diligent work. Overall, English use is adequate, and the authors provide a sound article structure, including the data supporting their findings. Background and literature references can be improved.

Experimental design

The research question is relevant and meaningful but can be redrafted to improve how it can fill gaps in acoustic monitoring literature. Methods are overall sound, although I identified some issues in the modelling approaches that should be reviewed.

Validity of the findings

Conclusions are sound, although the results and discussion can be improved for clarity.

Additional comments

This is an interesting topic for ecologists interested in acoustic monitoring. Emerging technologies are enabling to decrease research budget by the adoption of open-source alternatives that still provide high quality recordings. I must commend the diligent work by the authors in providing this manuscript, which is an important and timely contribution. Below, I raise general and line-by-line commentaries and suggestions for the authors to improve clarity and the strengthen of this work.

Introduction
I suggest the authors to improve two main points. First, the reasoning for testing SNR regarding sampling efficiency. How can wildlife research using acoustic monitoring can benefit from such approach? Second, the goals can be redrafted for clarity.

Methods
There is some missing information regarding the experimental design that should be reviewed. The number of replicates (e.g. 2 mic replicates per level) and other sources of temporal replication (e.g. monitoring day, source of templates in automatic detection) should be disclosed. Some concerns are raised regarding the suitability of the modeling procedures, which are detailed in the line-by-line suggestions. Additionally, it would be interesting to inform the reasons for choosing these microphone sensors in light of what is used in acoustic monitoring. This source of comparison can further be addressed in the discussion.

Results
Some missing reports are: i) model inference regarding transformed vs. non transformed values for SNR, ii) results of the modelling between sound pressure level ~ distances, iii) descriptive results about the registered species (e.g. list of species). Figure legends and axis labels can be improved.

Discussion
Overall, some points can be better discussed. First, I suggest the authors to include references to stand for their statements. Second, some points can be included for a more comprehensive discussion, such as i) how does SNR specified from mics relate to SNR from actual recordings, and ii) how can other settings improve SNR? For instance, gain level, height of deployment, etc…; iii) how similar are the SNR specifications of the investigated mics from those in commercial models (e.g. audiomoth, SM models…); iv) how can your findings support the choice of mic to be used in open-source platforms, such as Arduino?
Introduction

L.43: I suggest rephrasing for “for sonant animals as birds and bats).
L. 44: Please, include “digital” after “In a”.
L. 44-45: Consider rephrasing for: “Sound waves are transduced to electric signals by microphones and then converted in digital signal that…”.
L. 47: The authors could acknowledge that microphone quality is one of many variables that influence the detection of animal sounds (e.g. habitat structure, signaling behavior, climatic conditions, sample rate, quantization, among others).
L. 48: Please, include “active microphones”, since the specification of microphone SNR won’t apply for passive microphones (e.g. dynamic ones).
L. 48: For acoustic surveys, the frequency response of microphones and the pick-up field are also important components to characterize the quality of microphones.
L. 49: Please, state that the origin of self-noise is from the mic’s circuitry.
L. 62: The authors may consider explaining in more detail what they meant with “we will focus on the more commonly mentioned signal-to-noise ratios”.
L.66: Include “microphone specifications”.
L.79: To clarify the goals, the authors may consider redrafting this excerpt to first state their aims and then how they made it. For instance, the specific goals can be described as 1) test if observed SNR is consistent with manufactures’ specifications, 2) test if detection areas is influenced by SNR, and 3) determine the sample efficiency for bats and birds using manual and automatic methods of species detection in audio recordings.
L.79: Change for “12 types of microphones”.
L. 83-86: These sentences can be relocated to the methodological section.
L.87-89: It is true that acoustic sampling area can be estimated with detection ranges of standardized sound tests. However, the sound level from the emitter is largely variable between species and even within a vocalization. For instance, consider a species with a vocalization containing larger energy levels than the ones used to infer detection ranges. Whenever such species vocalize outside the estimated detection ranges, it can still be detected by microphones given its high amplitude levels. Therefore, the direct link assumed between SNR and measures of species diversity is arguable. I suggest the authors to rephrase and to reason the potential correlative relationship between SNR and measures of species diversity.

Methods

L. 104: Please, inform the intensity used in the calibration.
L. 135: Which statistical tests do the authors refer to? Up to now, none was mentioned.
L. 143: Change ‘recorders’ for ‘microphones’.
L. 154: Please, inform the sound levels used on the tests.
L. 155: Please, inform the frequency response of the loudspeaker.
L. 157: At which amplitude levels were the ultrasonic sequences emitted?
L. 167-174: The sequence of procedures is a bit difficult to follow. I suggest the authors to provide acronyms when explaining each step on the procedure explained previously and then cite them when explaining how to calculate SNR. For instance, give an acronym to “amplification applied to the recorder” when defining it previously, and repeat it here.
L. 168: I suggest using “sensors” or “microphones” instead “recorder”. Please, check consistency through the text.
L. 175-186: The type of fitted model is missing here. Which are the predictors and response variables? Also, the authors should state which is the goal of this analysis. Perhaps the authors should consider rethinking the need to apply such inference method to pursue how distinct microphones generate distinct SNR. Specially given a high number of treatment and small observations (12 levels, 2 per level). One suggestion is to visually explore such differences simply by plotting SNR (y-axis) by mic type (x-axis) and in one color the manufacture’s level, and the other colors the observed ones. This would also provide a clearer explanation in the results.
L. 193: The authors stated previously that up 10 meters, it was measured at 2-meter intervals (L. 160).
L. 195-198: So the authors excluded distances between 30-50 meters? Please, indicate whether this was necessary to estimate the extinction distances.
L. 199: Here it is also unclear how those models were fitted. Figure S1 suggests it was one model per microphone, which I believe that would not be proper way to test if distance affect SNR and if this decay differs between microphones.
L. 202: Please, include a description in figure S2.
L. 209-210: The authors may also check model residuals using transformed vs. non-transformed SNR values to check which is better suited for linear model.
L. 220-221: For how long?
L. 231: Check if figures S3 and S2 were shifted.
L. 233: These intervals are quite large… What’s the reason for it, since the authors state that recordings were synchronized?
L. 241: Why is bat richness not relevant? Also, change according L. 241 if bat richness was not measured.
L. 242: Include “call types”.
L. 242-243: Please, describe if the total duration of tagged vocalizations is the same as calculating call duration for each detected signal (and then describe how was call duration measured). Can you also provide a reference that used the same measure to characterize vocal activity?
L. 246: Please, describe the reason to test for differences in transformed vs. non transformed values.
L. 252: Please, indicate the time interval between each step.
L. 258: Please, indicate which bat call type was used.
L. 261: Please, indicate the analysis performed in monitor to detect calls.

Results

L. 273: Check the order of supplementary figures. Figure S4 seems to be unrelated to the test. Additionally, the authors should also provide a table with competing models and coefficient estimates.
L. 274-275: Does this information refers to the observed or the manufacturer’s specification?
L. 276: How did the authors extrapolated the extinction distances? It should be detailed in the methodological section.
L. 277-278: Perhaps this information can be moved to the methodological section.
L. 280: Explain why the models fitted in figure 1 are not linear (they are curved, perhaps indicating the use of a quadratic term). Additionally, provide a table with coefficient estimates. It is still unclear how many models were fitted.
L. 280: Which effect sizes does the authors refers to?
L. 285: Which method was used to estimate marginal R-squared?
L. 286: 360 seconds?
L. 286: Same commentary as the one in L.280 about the figure and effect sizes.
L. 289: This seems to be true for the first and last graphs. On the others, there seems to be an overlap of distinct SNR in higher species richness values. Additionally, the authors may provide a table with the results of the tests and coefficients.
L. 296: Figure 4 shows many points in the graph, indicating many observations provided by external sources. This is missing in the methodological section: the number of observations (n) used in each model. Moreover, is the model shown fitted in the black or both black and red values?
L. 297: 12 microphone types?
L. 298: These false-positives are certainly related to the type of analysis used to automatically detect calls. Ideally, the authors should provide information on the accuracy of the detector itself, monitoR is the one to blame. Therefore, the authors should consider removing the “inexplicably” and provide more information on the procedures used to automatically detect species calls. Moreover, I am concerned that a large proportion of false positives may be affecting the meaning of “proportion of correct calls”.
Figure 1 and 2. Please, include “log transformed” in the x-axis label.
Figure 3. As SNR is not a continuous variable in here, it should be depicted with a categorical color palette.

Discussion

L. 307-308: The authors can discuss a bit more this assertive.
L. 309-310: Unfortunately, the reader can’t see these differences without a proper figure. For instance, does mics with higher SNR always lead to higher observed SNR?
L. 348: Acoustic signature may be not suitable to describe the characteristic of a sensor. In fact, audio sensors have different frequency responses, that should be stated in the methodological section.
L. 382: You can also adjust this with gain level adjustment.

·

Basic reporting

Summary

This article is overall well written and presents interesting results for the field of ecoacoustic monitoring. The authors reveal that microphone SNR, which is rarely reported or considered in ecoacoustic sampling protocols, should be taken into account because it has a strong influence on sampling area around microphone and the proportion of animal vocalisations registered at one site. Quite logically, they find that the lower the signal ratio, the lower the area sampled. But having these estimations will be valuable to guide future sampling designs.

My main suggestions are mainly on the form. The methods are a bit hard to follow at times and lack crucial details, I suggest a thorough revision of the article. I have highlighted below some of the main points that were difficult to understand. I also suggest to include some of the supplementary materials back in the article to help the reader to follow along.

Introduction

An explanation of what sensitivity is and how it is measured is missing. Sensitivity appears in the methods but should be mentioned and defined earlier. Maybe to clarify the important variables (self-noise, SNR and sensitivity) and how they relate to each other, they could be defined in a dedicated box.

L56: remove ‘the’ in ‘as the 10 times’.
L57-61: this needs to be clarified. Maybe give the general formula? And give more details. Maybe switching the order of of the sentence L58 with the following two sentences could help.
L73: One thing that took me a bit of thinking is the fact that low SNR necessarily means high self noise. I thought other factors such as sensitivity of the microphone could factor in. I think that’s mainly because I hadn’t properly understood your definition of SNR when I first read it. Maybe it would be worth explaining a bit.
L75-76: isn’t that relationship obvious? Maybe you could rather emphasize that how much of an impact SNR has on signal detection in a natural setting has not much (never?) been quantified.
L79: what are the price ranges of the microphone elements tested? (I know they are discussed below) but do you know how much an assembled microphone would cost with these microphone elements? I can not believe that an assembled microphone can cost 4€.

Methods

The section call ‘sound transmission sequences’ is confusing as it aims at measuring sensitivity, SNR and detection spaces. Maybe it would help to explain the procedure for recording the sequences and then have a section for each of those three aims (or two sections but add details on how you calculate relative and absolute sensitivity and SNR).

L96: Remove ‘each of’
L99: Remove ‘of them’
L99: ‘used in commercial microphones’ what are the others used for?
L116: What does ‘pointing to the horizon’ mean? Isn’t that the same as parallel to the ground? Did you mean ‘horizontal’? At which height were they?
L121: SM2bat+ has a very particular audio plug that is usually only compatible with wildlife acoustics products. How is it possible that this was the only recorder that was compatible with all the microphones? Maybe reintroduce the supplementary materials in the methods and give more details…
L148-149: Why do you need sensitivity values? Explain.
L152: Sensitivity should have been defined before that.
L157: How long were the chirps?
L164: What is this test signal? Explain which variables you measure with it…
L172: How do you calculate self noise and SNR with this? Give more details. What was the calibrator used for?
L184: instead of ‘specified’, I suggest ‘manufacturer’ to clarify.
L185: What are these manufacturer variables?
L194-198: How do you get meter precision out of 5 meter measurements? As explaned here, it sounds like the experienced listener just eyeballed it. Maybe explain a bit more how the range was extrapolated for sensors that had ranges higher than 50 m (refer to figure S1?).
L200-202: How were the quarter ellipses drawn?
L206: Isn’t SNR already measure as a log? Why would it need to be transformed again? Technically if there’s only spreading, the relationship between SNR and detection space should be linear, shouldn’t it?
L207-209: Low/high values of what?
L225: In which software were the recordings screened?
L231: I think there is a confusion in the supplementary figure numbers: this is S4.
L258-262: It is not entirely clear what was used as a template and on which recordings were applied the detectors.

Results

I would be interested to see results about the consistency of measured SNR with this method, does it vary between manufacturers?
Why not report the self-noise measured and comment on them?
What are the relationships between sensitivity, self-noise and SNR?

L273: I don’t understand the link between the text and what I think is the figure you are referring to is S4.
L278: Could you explain the ‘terrain irregularities’?
L280-281 and L286-287: move to methods? This will avoid repetition.
L285: There were microphone elements for which there were not a single detection? That seems extreme!

Figures

Fig. 2: Which are the microphones that do not detect anything. Maybe they should be explicitly mentioned as bad for this type of application?
Fig. 4: Explain in the legend what the black line shows. Why not have two lines, one for when the template is internal vs external (black and red line).


Supplementary

Why not include more of the figures in the main text? For example the spectrograms (S4), and the estimated sampling areas (S2) as well as the figures showing measured SNR (S3 and S5)?
Are there legends for the supplementary figures?
S1: Make sure the order of the microphone is more “logical” (1, 2, 3…). And maybe use A, B, C… as in other figures. How come there isn’t 4 plots per microphone?
S2: Why are there only 12 circles? Were they averaged over each of the 2 element for each?
Figure S4: Were those the highest and lowest SNR sensors? If not, maybe show these?

Experimental design

The experimental design is appropriate but sometimes hard to follow (see Basic reporting for details). I suggest adding more details.

Validity of the findings

The findings are valid but some results that are set up in the introduction and methods (e.g. self noise) remain unreported even though they may be interesting.

Additional comments

I believe this article is a very valuable contribution to the field of ecoacoustics monitoring. Please make sure to clarify the methods.

---

## Round 0.2 · Minor Revisions

Reviewers have now returned their last minor comments on the manuscript. Like them, I believe the overall quality of the manuscript has considerably improved. Also, thank you for providing the R Markdown, which clarified many questions regarding data analysis.

R2, and to a lesser extent R1, pointed out a few minor adjustments that deserve to be implemented to improve the flow of the text and clarify some aspects ot the methods and results. I think you should keep the GLM without the interaction and not use a simple correlation. Please, include deltaAIC in your model selection table and a null model with only the intercept. So, I invite authors to make these last amendments and return their paper to a final decision. I believe the paper should be eventually published after incorporating these minor corrections.

·

Basic reporting

I am happy to see how much the methods section (the one that deserved more attention) and the manuscript has improved from the original submission. I am sure that it will be a very interesting ms for future developement of acoustic programmes.

I have no major comments. There are 2-3 things that I would have highlighted if the paper would have been focused on the utility of sound recorders for estimating bird richness or bird abundance, but as the main goa lis to compare how some acoustic variables differ among microphones with different SNR, I truly believe in the current results. I have just a few editing comments.

L. 59: Maybe “Finally, Bardelli et al. (2019) ….
L. 72: Maybe is better to say something like: “Our mail goal was to…” We aimed to”...
L. 100: Usually numbers lower than 10 are written with letters, not sure about the recommendations of PeerJ.
L. 173: It suggests that recorders were left only one night, but it was not the case. Please, specify the concrete number of days.
L. 251. Add that reference.
References1: Add article number to Knight ms. Idem for Turgeon, this journals uses article number instead of page numbers.
References2: Also check the reference to Katz in the text, sometimes is cited as et al and others as single author.
References 3: Idem for Darras et al. 2018, since there is 2018 but not 2018 a. However, there are 2018b and 2018c (not included in the reference list).
Maybe in the discussion you could valorate the possibility of say something about how using microphones with variable SNR may have a impact in the number of species detected using ARUs (I am not providing any reference since the authors already did a meta-analysis of the topic), but also in the number of birds estimated around recorder, since the detected vocal activity rate is a common method to estimate bird densities (or abundance from recordings).
For example:
Borker, A. L., McKown, M. W., Ackerman, J. T., Eagles‐Smith, C. A., Tershy, B. R., & Croll, D. A. (2014). Vocal activity as a low cost and scalable index of seabird colony size. Conservation biology, 28: 1100-1108.
Oppel, S., Hervias, S., Oliveira, N., Pipa, T., Silva, C., Geraldes, P., Goh, M., Immler, E., & McKown, M. (2014). Estimating population size of a nocturnal burrow-nesting seabird using acoustic monitoring and habitat mapping. Nature Conservation, 7: 1-13.
Pérez‐Granados, C., Bota, G., Giralt, D., Barrero, A., Gómez‐Catasús, J., Bustillo‐De la Rosa, D., & Traba, J. (2019). Vocal Activity Rate index: a useful method to infer terrestrial bird abundance with acoustic monitoring. Ibis, 161(4): 901-907.

Acknowledgements:
I think that it is worthy to thank the reviewers, since more or less they improved (at least a little) the ms. ;)

Experimental design

The methods section has been clearly improved, and even the most confusing section of the original version (to me), the one related to the external detection template, has been modified following a more common method, such as the estimation of the precision and the recall rate. By the way, please check the Knight et al. ms again and valorate if the term recall rate is more adequate, at least for me it is the adequate term (rather than recall).

Validity of the findings

Nothing to add. The impact, conclusions and data were already clear in the original submission, and have been even improved with the new R code and new analyses.

Additional comments

Thanks for your final words, and more importantly, for your work.

Cristian Pérez-Granados.

Reviewer 2 ·

Basic reporting

I have read the previous and the revised version of this manuscript, and I believe that most of the points raised by this reviewer were satisfactorily addressed. I think this is a much-improved manuscript that will be very helpful for the readers interested in ecoacoustics and acoustic monitoring in general, and I commend the authors for their diligent work.
I have raised some few concerns in this revised version of the manuscript. Some are related to few points that haven’t changed from the previous version. I truly respect the decision by the authors, but in my viewpoint, some minor adjustments would improve the understanding of this important work.
Particularly, the additional concerns raised in this review regard to some procedures in the statistical analyses, which I detail below. Additionally, some of the missing information in the main text were justified to be in a report (markdown file) from the supplementary material. I do applaud the authors for sharing the code and preparing the report, but I believe that this is rather an auxiliary material that do not substitute synthesized results in the form of tables. Therefore, I still recommend the authors to include some main information in the text and to provide tables with results summaries.

Abstract/results,
• Change “from worst to the best” for “from the lower to higher SNR”. Judging what’s worst or best should be left for the reader.

Introduction
• L. 42: “…microphone quality is essential as it is a controllable technical parameter…”.
Although I understand the message here, I believe the microphone quality is fixed rather than controllable, particularly if by microphone quality you mean the self-noise, as stated on the next sentence.
Things that can be controlled are the technical settings and recording techniques in the field. And last but not least, when customizing recording systems, the choice of microphone is controllable as well. That being said, I suggest the authors to first define what they mean by microphone quality, and then justify that the self-noise can affect sound detections.

Methods
• Measured vs. specified signal-to-noise (L. 209)
I am unsure if glm is the most appropriate method to test the association between these variables, since there is no clear directional cause in here, but rather, a correlational one. However, I do like the idea of including the manufacturer in the model, although I think they should not be in the form of interaction (*). To complement these analyses, the authors could additionally include a correlation test among the variables.
• For model inference with AIC, the authors should also include a null model so we can check if the relationship is different from that expected by chance alone. This model is built as a function of intercept (e.g. null.model <- lm(response ~1)). Additionally, I recommend to also provide the differences between AIC of the best model and each AIC (i.e. ΔAICi, which can be computed with ICtab() in bbmle package).

• L. 199: As suggested in the previous revision, I believe that the type of automatic detections made in monitoR should be disclosed here, and the authors can add that details can be found in the supplementary material.

• Although we are all sure that SNR will affect somehow detection space area, vocalization activity, species richness, and automatic detector performance, other mics’ specifications could also potentially act as confounding effects. Specifically, many potential confounding effects could be summarized by attributing the effect of being from different manufacturers.
However, I believe the number of observations is low to properly include manufacturer as random effects in the models. Therefore, a solution could be to build models including manufacturer as covariates and then confront the different models (e.g. using AICc).

Results
• As required in the previous revision, the authors should describe their results using tables containing the predictor variables, weighted AIC, degrees of freedom, etc.

• In figure 3, the values for birds were predicted from a model using log(SNR). Shouldn’t the values be subject to exp(), just as it was done with bats?

• I understand that SNR is a continuous variable. Let me rephrase the suggestion, which is just a way to “read” more easily the figure and rapidly understand what it means. The observed SNR values range between 20 to 80 (1kHz). The color scale used is an even distribution of gray levels. However, SNR values observed they are not evenly distributed along the range of these values. Specifically, differences between close values of SNR are nearly impossible to distinguish with the color gradient used. Therefore, the authors could use a color gradient that emphasizes such minor differences between the aggregated values.

Experimental design

No comment.

Validity of the findings

No comment.

·

Basic reporting

The authors have significantly improved the manuscript according to all the reviewers and editor’s comments.

Experimental design

The experimental design is adequate considering the questions addressed.

Validity of the findings

The findings are clearly support by results and now better explained.

Additional comments

The findings are clearly support by results and now better explained.

---

## Round 0.3 · accepted · Accept

Thank you for clarifying these last minor concerns. I’m happy to recommend it for publication.